# Autonomy of Nursing Students: Methodological Study of Validation of the PALOP Scale Portuguese Short Version

**DOI:** 10.3390/ijerph20217014

**Published:** 2023-11-02

**Authors:** Luís Manuel Cunha Batalha, Josefa Palop-Muñoz, Carlos Alberto Cruz de Oliveira, Carlos Saus-Ortega, Paulo Alexandre Carvalho Ferreira, María-Rosario Gómez-Romero

**Affiliations:** 1Health Sciences Research Unit: Nursing (UICISA: E), Nursing School of Coimbra (ESEnfC), 3000-076 Coimbra, Portugal; oliveira@esenfc.pt (C.A.C.d.O.); palex@esenfc.pt (P.A.C.F.); 2Nursing School La Fe, University of Valencia, 46026 Valencia, Spain; josefapalop@gmail.com (J.P.-M.); saus_car@gva.es (C.S.-O.); gomez_ros@gva.es (M.-R.G.-R.)

**Keywords:** nursing, professional autonomy, nursing students

## Abstract

The cultivation of critical thinking and decision-making skills promotes student autonomy. Only a few instruments measure nursing students’ autonomy, and the PALOP^®^ Scale is one of them. This study aimed to semantically and culturally adapt the PALOP^®^ Scale to European Portuguese and assess the psychometric properties of a short version. A methodological study was conducted with 530 second and fourth-year undergraduate nursing students. Content validity was assessed using exploratory and discriminant factor analysis, and reliability was determined through analyses of internal consistency, temporal stability, and floor and ceiling effects. The analysis of the psychometric properties of a short version of the PALOP^®^—PT Scale revealed complete agreement (100%) among panel members for content validity. The scale also showed discriminative capacity among second- and fourth-year students (*t* (528) = −7.907, *p* < 0.001) with a five-factor structure, with a total explained variance of 57.2%. Reliability analysis showed excellent internal consistency (*α* = 0.935) and moderate temporal stability (95% ICC (3.1) = 0.520 [0.290—0.693], *p* < 0.01). The short version of the PALOP^®^–PT Scale is a promising tool to assess nursing students’ perceived autonomy and identify necessary adjustments to their professional identity.

## 1. Introduction

The concept of autonomy is polysemous and varies according to context and personal characteristics. It encompasses philosophical, theological, psychological, ethical, legal, and organizational dimensions [1]. Although these dimensions follow similar criteria and values, they may appear contradictory [1,2]. Autonomy should not be confused with self-reliance or independence, as they emphasize individual decision making or empowerment, particularly in power relations [2]. Broadly speaking, autonomy refers to the capacity for self-government, that is, owning oneself [2].

Comprehensive care within multiprofessional teams requires extensive and close collaboration among team members within different areas of expertise. In order to prioritize the comprehensive care model, health professionals’ training should promote their capacity for collective decision making while respecting their professional autonomy. Modern curricular models support reformatory approaches aimed at greater inclusion, emancipation, and social justice. Pedagogical theories have greatly influenced nursing education since the 1980s, with autonomy as one of the primary goals. Educators promote student autonomy by encouraging critical thinking and enhancing decision-making skills [2].

It is the responsibility of all involved to connect theory and practice. Academics play an important role in creating favorable conditions for nurses to be autonomous in their profession. To achieve this, students must internalize a culture of critical thinking and engage in practical exercises in clinical teaching contexts.

Considering the rising complexity of healthcare systems, it is worth examining whether nursing education adequately prepares students to become autonomous practitioners and to assume new professional roles. The concept of student autonomy includes key dimensions such as responsibility, confidence in skills (self-efficacy), engagement with student-led learning (self-regulation), and depth of knowledge [3]. This vision is consistent with the mission of higher education, which values and promotes students’ active role in learning.

There are two schools of thought on student autonomy [3]. One values individual characteristics and is not very changeable, and the other is linked to successful learning experiences and the outcomes of comprehensive learning, usually culminating in student-led research projects, and therefore is changeable and can be developed over time [2,3]. Assuming that autonomy can be promoted through specific teaching strategies (clinical placement, internships, research work, autonomous group learning activities, and workshops) and self-assessment [2,4], it is essential to assess how student autonomy develops during their training.

Autonomous action involves the responsible use of freedom toward oneself, others, and society. Autonomy is learned when an individual gains the ability to establish and carry out self-determined rules and functions [5]. Nursing students’ autonomy is their ability to act independently, determine their own conduct in care provision without external influences, and become increasingly participatory while taking responsibility for their actions [6].

Autonomous nursing is closely linked to developing a professional identity that reflects the profession’s unique social value. This involves building a body of knowledge specific to the nursing profession [7], which is an indicator of the development of autonomy. This development strengthens schools’ commitment to their integral education and promotes student engagement in learning, research, and pedagogical practice, developing their decision-making capacity and responsibility [8]. These skills are crucial for nurses to tackle new, ambiguous, and uncertain situations in the future [9]. There is limited knowledge about nursing students’ autonomy, how it progresses during their studies, and how it is measured and evaluated as a component of professional identity [5]. Moreover, there are only a few instruments for measuring nursing students’ autonomy [3] and no instrument for assessing Portuguese nursing students’ autonomy.

This study aimed to provide a practical tool for assessing the autonomy of Portuguese nursing students. The PALOP^®^ Scale was developed in Spain to assess nursing students’ autonomy. The decision to use this scale in Portuguese settings is justified by the cultural, professional, and educational similarities between Spanish and Portuguese nurses.

## 2. Materials and Methods

### 2.1. Study Design

In order to semantically and culturally adapt the PALOP^®^ Scale to European Portuguese [1] and assess the psychometric properties of a short version, a methodological study was conducted to translate, semantically and culturally adapt, and assess the psychometric properties of a short version for nursing students, following the recommendations of Sousa and Rojjanasrirat [10].

### 2.2. Ethical Considerations

Data collection was approved by the Management of a Nursing School and the Ethics Committee of the Health Sciences Research Unit: Nursing (Opinion 578/04-2019).

### 2.3. Participants and Setting

The population consisted of second- and fourth-year nursing students attending theoretical and/or theoretical–practical classes at the time of data collection. Participation was voluntary and anonymous. Of a total of 628 students, 530 were included in the final sample, representing 84.4% of the total number of students. All participants had undergone at least one clinical teaching experience in either a hospital or community setting.

### 2.4. Adaptation Procedure

The study was carried out in six phases: translation into European Portuguese, comparison and synthesis I, back-translation, comparison and synthesis II, pilot test of the Portuguese version, and psychometric analysis.

In Phase 1, the author authorized the translation of the original instrument into European Portuguese by two independent bilingual translators whose first language was Portuguese.

In Phase 2, the two translated versions were compared with the original version by a third bilingual translator. A panel including the third translator, the two translators, and three research team members analyzed any discrepancies and reached a consensus on a synthesized translated version (synthesis I).

During Phase 3, two bilingual translators whose mother tongue was Spanish independently back-translated the synthesized version.

In Phase 4, the two back-translations were compared to each other and to the original instrument by a panel of nine members, including four translators involved in the translation and back-translation process and five research team members (a methodologist, the original scale’s author, and three other nursing teachers). This step aimed to assess the conceptual, semantic, and content equivalence of the back-translated instrument and build the Portuguese version (synthesis II). Decisions were made if seven or more of the nine panel members agreed.

Phase 5 consisted of a pilot study. The preliminary Portuguese version was applied in three runs to three nursing students to assess the instructions, the items, and the response format.

In Phase 6, the psychometric properties of the Portuguese version were analyzed in a sample of 530 second- and fourth-year nursing students.

After the psychometric analysis, 47 items were removed, including 10 from the personal characteristics dimension and 37 with floor and ceiling effects. As a result, the PALOP^®^ scale was reduced to 26 items, referred to as PALOP^®^—PT reduzida (PALOP^®^—PT—short version).

A seven-member panel, including three Portuguese and four Spanish nursing teachers (one of whom was the original scale’s author), assessed the content validity of the items in this short version. All panel members agreed to maintain the items.

The psychometric properties of this version were assessed for content validity using exploratory and discriminant factor analyses and for reliability using internal consistency, temporal stability, and floor and ceiling effects.

### 2.5. PALOP^®^ Scale

The PALOP^®^ Scale assesses nursing students’ perceived autonomy [1] using 73 items divided into 10 dimensions: personal characteristics (10 items), helping relationship (7 items), individualized care (8 items), interdisciplinary dialogue (5 items), professional development (4 items), responsibility (8 items), excellence of care (8 items), organization and communication (6 items), application of nursing knowledge (8 items), skill mastery (5 items), and freedom and limits (4 items). Each item is rated on a Likert-type scale from 1—*strongly disagree*; 2—*disagree*; 3—*neither agree nor disagree*; and 4—*agree* to 5—*strongly agree*. The total score ranges from 73 to 365 points.

The scale was validated in a sample of 26 third-year nursing students who had completed at least one clinical placement within hospital and community settings. The decision to select third-year students was practical because they were available at the time to participate. Exploratory factor analysis of a 63-item version (excluding 10 items that had *r* < 0.2 with the total scale) revealed a single-factor structure with an explained variance of 30.1%. The scale also had high internal consistency, with a Cronbach’s alpha of 0.939.

### 2.6. Data Analysis

Data were analyzed using IBM SPSS Statistics for Windows, version 27.0.

Descriptive analysis for sample characterization used absolute and percentage frequencies for nominal data and measures of central tendency such as mean and standard deviation for continuous data.

Exploratory and discriminant factor analyses were conducted to determine content validity. Factorial validity of the domains was determined using the Kaiser–Meyer–Olkin (KMO) test with acceptable values above 0.6, Bartlett’s test of sphericity with acceptable values of X^2^ for *p* < 0.001, and Varimax factor rotation with acceptable values above 0.6 [11] A factor loading greater than 0.30 was considered significant. Discriminant validity was determined using Student’s *t*-test for independent samples between second- and fourth-year students.

Reliability was assessed by calculating internal consistency, temporal stability with a 30-day interval, and floor and ceiling effects. The Kuder–Richardson formula was used to calculate internal consistency based on Cronbach’s alpha values. Alpha values of ≥0.5 were considered poor, ≥0.6 questionable, ≥0.7 acceptable, ≥0.8 good, and ≥0.9 excellent [12]. Temporal stability was assessed using the intraclass correlation coefficient (ICC) with a 95% confidence interval for two-way, mixed-effects, absolute agreement, single-rater/measurement model. Interpretation was rated poor if less than 0.5, moderate if less than 0.75, good if less than 0.9, and excellent if greater than or equal to 0.9 [13]. Values above 20% were considered poor reliability indicators for floor and ceiling effects (lack of discriminatory power) [14].

## 3. Results

### 3.1. Sample Characterization

A total of 530 students participated in the study, of whom 464 (87.5%) were women. They had a mean age of 21.5 ± 2.7 years, ranging from 19 to 42 years. A total of 459 (86.6%) students had the course as their first choice; 505 (95.6%) had completed 12 years of schooling before applying, 18 (3.4%) already had another degree, and 5 (0.9%) had another qualification. Most students (504; 95.1%) did not work, and 470 (88.7%) had never worked before starting the course. The nursing students were either in the second (230; 43.4%) or fourth (300; 56.6%) years. The mean age of second- and fourth-year students was 20.4 ± 2.0 and 22.3 ± 2.9 years, respectively. Most second-year (87.0%) and fourth-year (88.0%) students were women. The course was the first choice of most second-year (85.2%) and fourth-year students (87.7%). Most second-year (95.2%) and fourth-year (95.3%) students had completed 12 years of schooling. Few second-year (2.6%) students worked, while the percentage of fourth-year working students was higher (6.7%). Few students had work experience before taking the course: 8.7% (20) in the second year and 13.3% (40) in the fourth year.

### 3.2. Adaptation Procedure

The translation and cultural adaptation of the scale into European Portuguese went through five phases and achieved a high degree of consensus among the participants. The few divergences concerned the terms “desacordo”, “acuerdo”, “paciente”, “ordem”, “transmisiones”, “prejuízo”, “argumentados”, “elejo”, and “aproveito”, which were, respectively, translated into “discordo”, “concordo”, ”pessoa”, ”atualizada”, “comunicações”, “preconceito”, “fundamentados”, “escolho”, and “assumo”. No divergences were found between the original version and the two back-translations. The concept, semantic, and content equivalence of the Portuguese version was determined based on a 99% agreement among the nine panel members. Only one of the nine members disagreed on items 1, 8, 13, 30, 72, and 73.

The clarity of the instructions, items, and response format was assessed by applying the Portuguese version to three nursing students in three runs. In the first run, all items were rated clear except for items 4, 8, 17, 18, 20, 23, 56, and 58, which were rated unclear in one response. In the second run, all items received full agreement except for item 4, which did not receive full agreement until the third run. It was suggested that the phrase “common sense” be replaced with “clear language” to improve this item.

### 3.3. Psychometric Properties of the PALOP^®^—PT Scale

Factor analysis of the items of the PALOP^®^—PT Scale was performed to measure their psychometric properties in terms of content validity for item selection and deletion. The analysis revealed good values for exploratory factor analysis with KMO = 0.958 and Bartlett’s test of sphericity with *p* < 0.001. The Varimax rotation test yielded 15 factors with a total explained variance of 61.3%. Eight items (2, 4, 9, 19, 34, 41, 42, and 72) had factor loadings greater than 0.7.

Internal consistency assessed reliability, which was high, with a Cronbach’s alpha of 0.963. Corrected item-total correlations ranged from 0.21 (item 9) to 0.694 (item 40), whereas deletion of an item yielded a range of values from 0.964 (item 4) to 0.962 (multiple items). Values for the dimensions ranged from 0.654 for Personal characteristics to 0.875 for responsibility (Table 1).

Regarding temporal stability, the total scale showed a good level of agreement, with 95% ICC (3.2) = 0.791 (0.638–0.879), while most dimensions revealed a moderate level of agreement. The dimensions “Skill mastery” (95% ICC (3.2) = 0.432, 0.016–0.672) and “Individualized care” (95% ICC (3.2) = 0.454, 0.070–0.681) exhibited weak agreement.

A ceiling effect (strongly agree) was observed in 37 items (10 to 21, 24, 25, 31 to 42, 44 to 46, 48 to 50, 52, 56, 66, 70, 71, and 73), whereas only 1 item (9) showed a floor effect (strongly disagree).

Of the 73 items in the PALOP^®^—PT Scale, 47 were eliminated (10 items related to Personal characteristics and 37 showing ceiling and floor effects), resulting in a 26-item scale (PALOP^®^—PT Scale—short version).

### 3.4. Psychometric Properties of the PALOP^®^—PT Scale—Short Version

The psychometric properties of this short version were analyzed for content validity, revealing full agreement (100%) among the panel members.

The exploratory factor analysis was supported by KMO values of 0.951 and Bartlett’s test of sphericity with *p* < 0.001, indicating a five-factor structure that accounts for 57.2% of the variance. Fourteen items (22, 23, 26, 27, 28, 46, 57, 58, 60, 61, 65, 66, 69, and 72) demonstrated a factor loading greater than 0.6, with the lowest being 0.372 (item 43) (Table 2).

Regarding discriminant validity, second-year students obtained lower mean scores than fourth-year students (98.1 ± 10.4 vs. 105.0 ± 9.8). This difference was statistically significant (*t*_(528)_ = −7.907, *p* < 0.001).

The scale showed high internal consistency, with a Cronbach’s alpha of 0.935. The alpha values ranged from 0.701 (Factor 4) to 0.851 (Factor 2) (Table 3). The scale also had moderate temporal stability, with a 95% ICC of 0.520 (0.290–0.693), *p* < 0.01.

## 4. Discussion

The scale was adapted to European Portuguese based on internationally accepted procedures for semantic and cultural adaptation. Because of similarities between the Spanish and Portuguese languages and cultures, a high degree of consensus was achieved in the translation, semantic, and cultural adaptation stages. The panel’s decisions were supported by the original scale’s author, who played a crucial role in clarifying the semantic and cultural adaptation of some items.

The pilot test indicated that nursing students understood the items and that the instructions for completing the scale were clear. However, students found completing the scale somewhat lengthy and uncomfortable. This discovery highlights the importance of creating a short version to assess student autonomy without losing information.

The PALOP^®^—PT Scale was analyzed for its psychometric properties and resulted in a 15-factor structure and a much higher explained variance than the original scale, which has only a single factor with a lower explained variance [1]. The results in the original version are mainly justified by the small sample size involved in its validation.

The analysis of the validity and quality of the items revealed that few contributed little to measuring the student’s perceived autonomy.

Hence, a shorter version of the scale was built. By deleting non-discriminatory items, including those related to personal characteristics, over half of the original items had to be deleted. Since student autonomy is a changeable characteristic linked to learning experiences, items that assess autonomy but are not susceptible to change over time or are unaffected by the quality or type of teaching provided, such as students’ personal characteristics, should not be retained [2,3].

The scale showed excellent internal consistency, but the consistency of the “personal characteristics” dimension was questionable [12], indicating the need to eliminate its items.

The good temporal stability [13] is in line with what is expected for a 30-day interval between applications.

The PALOP^®^—PT Scale—short version showed adequate properties to assess nursing students’ perceived autonomy. It is a promising tool because it is simple, easily accessible, and quickly applicable. During the validation phase, all items received consensus. The five-factor structure values safety, self-efficacy, mastery of profession-specific skills related to individualized care, integration into a healthcare team, and students’ active learning engagement.

Factor 1 includes items related to safety and confidence in one’s abilities (self-efficacy). Factor 2 pertains to profession-specific competencies, including knowledge, skills, and attitudes consistent with the professional identity. Factor 3 is associated with self-regulation and learning engagement. Factor 4 relates to care design, such as defining diagnoses, establishing outcome criteria, and prescribing interventions. Factor 5 prioritizes individualized care and considers the uniqueness of each person in the care process. Factors 4 and 5 could be combined into one dimension since they both relate to profession-specific competencies, namely the design of individualized care within a healthcare team. The total explained variance is satisfactory and significantly higher than the original version [1].

The PALOP^®^—PT Scale—short version includes key dimensions of nursing student autonomy, including mastery of profession-specific competencies and perceived self-efficacy, which is the first step in developing autonomy [15,16].

Student autonomy is a skill that can be improved throughout their studies [3]. Therefore, it was expected that fourth-year students would show a higher level of autonomy, which was confirmed. Even though this difference is statistically significant, a total mean difference of over seven points was expected. Nonetheless, it is consistent with the teaching context, as fourth-year students have more clinical experience. Further studies should examine the effectiveness of this short version in terms of its discriminatory power and its correlation with teaching methods that promote student autonomy, such as problem-based learning [4].

The availability of this scale to assess students’ autonomy during their studies allows for rapid curriculum intervention by using autonomy-friendly pedagogical practices. Unperceived progress in autonomy can be a significant barrier to student development [3]. In addition, the use of a standardized instrument allows for data comparison across studies.

This short version is reliable, as demonstrated by its excellent and acceptable internal consistency for Factors 4 and 5 [12] and good temporal stability [13]. The improved temporal stability of the short version compared to the 73-item version is a positive indicator. This is because student autonomy is not expected to change significantly over a short period of time. In addition, the lack of items with a floor and ceiling effect indicates that the short scale is reliable and discriminatory.

The PALOP^®^—PT Scale—short version consists of 26 items that assess nursing students’ autonomy on a scale with a minimum score of 26 and a maximum score of 130. To facilitate the analysis, this scale should be converted into a percentage by using the formula (X − 26)/10,400, where X represents the total score obtained.

The availability of a Portuguese instrument with good psychometric properties to assess nursing students’ autonomy allows for the development of curricula that use teaching methods for improving nursing students’ autonomy. It also allows for an evaluation of the actual value of teaching strategies that guarantee this desirable outcome.

To ensure that the nursing profession has greater autonomy, students’ autonomy throughout their studies should be assessed, and any necessary pedagogical adjustments should be made to the curricula.

Identifying the factors that improve students’ autonomy throughout their studies and clinical practice facilitates the development and improvement of strategies that promote critical thinking and autonomy, thus clarifying their professional identity and adding value to society.

### Limitations

A limitation of this study was that it was conducted at a single nursing school despite the large sample size. Another was that the factors affecting student autonomy, such as learning success or failure, were not assessed, which may hinder the interpretation of the results. The analysis of the scale’s discriminatory power and criterion validity is a limitation that future studies should address.

## 5. Conclusions

The semantic and cultural adaptation of the PALOP^®^ scale to European Portuguese was performed without problems. Based on the psychometric properties of the Portuguese version, a short version was developed that is quick to complete and easy to understand, has high discriminatory capacity, excellent internal consistency, and moderate temporal stability.

This shorter scale has a five-factor structure with an acceptable total explained variance. It assesses nursing students’ autonomy and values safety and self-efficacy, mastery of profession-specific competencies linked to individualized care design, integration into a healthcare team, and students’ active learning engagement. In addition, the scale exhibits discriminatory power, reliable internal consistency, and moderate temporal stability.

Further studies are needed to examine the scale’s criterion validity and discriminatory power with students in other nursing schools. This scale is a promising tool for assessing nursing students’ autonomy and monitoring it throughout their learning process. In clinical nursing practice, the scale will help identify the necessary changes to educational strategies that promote critical thinking and full autonomy, leading to a clarified professional identity and recognition of the social value of nursing.

Student autonomy is developed in decision making and the delivery of holistic and individualized care, promoting agency that goes beyond standard practice, resulting in improved quality of care. Lifelong learning, recognition of limitations, confidence in their knowledge, and successful learning foster in nursing students a sense of appreciation for their work and for nursing as a profession.

Nurses’ roles are constantly changing. Educators must anticipate these changes to promote students’ professional autonomy through the development of skill-based curricula that support autonomy using pedagogical strategies such as problem solving and group research.

## Figures and Tables

**Table 1 ijerph-20-07014-t001:** Internal consistency of the PALOP^®^—PT Scale.

PALOP^®^—PT Scale	Cronbach’s Alpha	If Deleted
Total	0.963	73
Personal characteristics	0.654	10
Helping relationship	0.800	7
Individualized care	0.857	8
Interdisciplinary dialogue	0.828	5
Professional development	0.837	4
Responsibility	0.876	8
Excellence of care	0.833	8
Organization and communication	0.789	6
Application of nursing knowledge	0.843	8
Skill mastery	0.824	5
Freedom and limits	0.758	4

**Table 2 ijerph-20-07014-t002:** Distribution of factor loadings of items across factors.

PALOP^®^ PT Scale—Short Version	Item	Factors				
		1	2	3	4	5
I decode the information of the person and their environment to understand its meaning and use it in the care process	17	0.310				0.563
I set goals to enhance the person’s abilities, adapting them to their expectations and available health resources	22					0.743
I plan actions to respond to people’s health needs using the available resources and following the organization’s policy	23					0.647
I trust myself when it comes to performing a technique (e.g., blood draw)	26	0.716				
I feel secure in my learning/knowledge and share it with the professionals in the unit	27	0.780				
I believe I can provide relevant information to the rest of the team	28	0.730				
I make reasoned professional judgments	29	0.546				
I establish a collaborative relationship with the work team	30	0.404	0.437			0.308
I help the person make decisions, from psychosocial skills to personal coherence	43			0.372	0.339	0.365
I identify the factors that can interfere with my professional judgments	46			0.679		0.308
I implement the appropriate strategies during nursing care, taking into account the person’s, the family’s, and my physical, historical, cultural context	48		0.388	0.537		
I support my observations with the evidence	50	0.455		0.411		
When I take care of a person, I know the pathophysiology that determines their clinical situation	57			0.662		
I use the different valuation systems with skill and dexterity	58			0.617		
I recognize and relate the signs and symptoms in the different pathologies	59	0.326		0.482	0.476	
I identify health problems and complications and am able to develop a nursing diagnosis	60				0.675	
I am able to determine the criteria of the results for each of the diagnosed problems	61			0.334	0.683	
I am able to explain and argue the criteria I used to make a clinical judgment	62	0.489			0.442	
I recognize the therapeutic and adverse effects of drugs and other therapies	63	0.451	0.303			
I identify a person’s physical and emotional responses	64		0.482	0.415		
I master the essential concepts of the nursing discipline: person, health, environment, and nursing care	65		0.743			
I reflect on and critically analyze the actions performed	66		0.651			
I am able to apply technical skills, using intuition and clinical judgment	67	0.315	0.559		0.318	
I distinguish and appropriately apply protocolized and non-protocolized care	68		0.548	0.316		
I recognize the essential concepts of the nursing discipline in decision-making in care planning	69		0.694			
I act freely and choose the behaviors that seem most appropriate to me in the context and situation at hand	72				0.629	
**Total explained variance**	57.2					
		13.9	12.9	11.8	10.1	8.5

Factor loading > 0.30. Principal component factor analysis with Varimax rotation. Total explained variance of 57.2%.

**Table 3 ijerph-20-07014-t003:** Internal consistency of the PALOP^®^—PT Scale—short version.

PALOP^®^ PT Scale—Short Version	Items	Cronbach’s Alpha
Total		0.935
Factor 1	26, 27, 28, 29, 62, 63	0.817
Factor 2	30, 64, 65, 66, 67, 68, 69	0.851
Factor 3	43, 46, 48, 50, 57, 58, 59	0.822
Factor 4	60, 61, 72	0.701
Factor 5	17, 22, 23	0.727

## Data Availability

The data presented in this study are openly available in https://esenfc-my.sharepoint.com/:f:/g/personal/batalha_esenfc_pt/Enk-tibpNSlNhkuyP1soVOsBE5Z8rzF_vXh68cRB19SVuA?e=HCgLPd (accessed on 21 August 2023).

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
