# Peer review of "Autonomy of Nursing Students: Methodological Study of Validation of the PALOP Scale Portuguese Short Version"

_ijerph, 2023, doi:10.3390/ijerph20217014_

Round 1

Reviewer 1 Report

Comments and Suggestions for Authors

The study objective is important, and the findings contribute to nursing education.

Two sources are too dated (10,12)

In item 8 in the list of sources, the year of publication is not specified.

In my opinion, it is important that Table 2 in the article will be written in English.

Good luck!

Author Response

Dear Reviewer,

Thank you very much for taking the time to review this manuscript. Find the detailed answers below and the corresponding revisions/corrections marked in red (attached file).

Best regards,
Luís Batalha

Reviewer 2 Report

Comments and Suggestions for Authors

The publication is valuable, and the validated tool may be interesting for other research. Unfortunately, I do not speak Spanish or Portuguese and it is difficult to understand what variables were analyzed in the publication. A good practice would be to translate it into an international language, e.g. English. Additionally, the structure of the summary requires correction in accordance with the standards applicable in this area. Keywords from the article are not from MeSH;
The introduction lacks the purpose of the work. A validated tool that may be part of the goal is listed. Without a goal, it is difficult to evaluate the effect achieved in a publication; It was explained why it was decided to use the tool.
The methodology should indicate inclusion and exclusion criteria in the description of the sample.
The tool was used in a shortened version, so can it be compared with the results of studies from Spain?
The translation process was described in great detail.
117-125 description of how the survey was conducted among 530 students; 129-141 - description of tool validation for 29 students. What was the first study that eliminated some of the variables, or the validation of the tool. What is the purpose of the description from lines 117-125 if the results contain a similar description.
Author Contribution is illegible
Conclusions should relate to the goal.

Author Response

(The authors gave the same response as above.)

Reviewer 3 Report

Comments and Suggestions for Authors

Dear Authors,

your work provides valuable insights into student autonomy in nursing education, especially within a Portuguese context. With a few refinements, your study has the potential to make a significant contribution to the field. Well done and best of luck with the next stages of your publication journey!

My suggestions are as follow:

Introduction Section: While the introduction provides context, it would be beneficial to emphasize why student autonomy in nursing is vital and its implications for patient care and the broader healthcare system. Adding a few references to reinforce the importance of this subject would also enhance its depth.

Materials and Methods Section: The research design appears appropriate. However, a more detailed description of the methods, especially on the selection criteria for participants and any biases accounted for, would offer clarity and enhance reproducibility.

Results are well presented.

Discussion is well written.

Conclusion Section: Your conclusions aptly encapsulate your findings. It might be worth mentioning potential implications for future nursing practices or pedagogical changes to underline the importance of your study. It would be beneficial to include specific recommendations for educators or curriculum developers based on your findings or you could propose concrete steps or strategies to incorporate autonomy-building pedagogical methods in nursing education.

Author Response

(The authors gave the same response as above.)
